# Marine carbonate system variability from tidal to seasonal timescales at the interface between the North Sea and Wadden Sea

Yasmina Ourradi<sup>1</sup>, Gert-Jan Reichart<sup>1,2</sup>, Sonja van Leeuwen<sup>3</sup>, Matthew P. Humphreys<sup>1</sup>

- Department of Ocean Systems (OCS), NIOZ Royal Netherlands Institute for Sea Research, P.O. Box 59, 1790 AB Den Burg (Texel), the Netherlands
  - <sup>2</sup>Department of Earth Sciences, Utrecht University, Utrecht, the Netherlands
  - <sup>3</sup>Department of Coastal Systems (COS), NIOZ Royal Netherlands Institute for Sea Research, P.O. Box 59, 1790 AB Den Burg (Texel), the Netherlands

Correspondence to: Yasmina Ourradi (yasmina.ourradi@nioz.nl)

Abstract. Shelf seas are important for global carbon cycling, but their carbonate system dynamics remain poorly understood due to complex spatial and temporal variability driven by interacting biological, physical and hydrological processes. To understand these complex dynamics, we focused on the Marsdiep channel at the Wadden Sea-North Sea interface, where strong tidal exchange creates ideal conditions for investigating carbonate system variability across multiple timescales. Highresolution (10-min) measurements of pH, combined with discrete sampling of dissolved inorganic carbon (DIC), total alkalinity (TA) and pH, were conducted over approximately one year between February 2022 and January 2023 at this location. We developed a multi-linear regression (MLR) model based on salinity and tidal data to predict TA (TA<sub>pred</sub>) (RMSD = 17.5  $\mu$ mol kg<sup>-1</sup>), and subsequently calculated DIC from the TA<sub>pred</sub> and pH (RMSD = 19.8  $\pm$  1.9  $\mu$ mol kg<sup>-1</sup>). To evaluate the performance and long-term stability of the pH sensor, we applied a semivariogram approach - an approach not commonly used in this context - to estimate pH sensor uncertainty and drift patterns over time. We propose this method as a robust approach for assessing sensors performance in short and long-term deployments at sea, particularly when calibration sampling frequency is irregular. Results revealed pronounced diel and seasonal variability in pH (seasonal range: 0.6), DIC (419 µmol kg<sup>-1</sup>), and TA (213.7 µmol kg<sup>-1</sup>). These fluctuations reflected the interplay of biological and hydrological processes, with pH mainly controlled by biological process, TA by hydrological processes, and DIC influenced by the combination of both. We used the Wasserstein distance to quantify the balance of processes driving DIC at any given time. During this study period, the Wadden Sea acted as a net CO<sub>2</sub> source to the atmosphere, with an annual release of 4.7 g-C m<sup>-2</sup>. A deeper understanding of the influence of biological and hydrological controls on the marine carbonate system is still needed to unravel the relative importance of the processes involved, especially in regions of higher complexities such as the Wadden Sea. Continuous high-frequency measurements may provide a tool to capture these dynamics across multiple time scales, from hourly to seasonal and interannual, in order to refine our understanding of their role in driving the carbonate system and in regional carbon cycling under changing climatic and hydrological conditions.

#### 1 Introduction

35

45

Despite covering only 7% of the global ocean surface, shelf seas are estimated to account for approximately 10 to 22% of global marine CO<sub>2</sub> uptake (Borges, 2011; Le Quéré et al., 2018; Regnier et al., 2013). These ecosystems play a crucial role in carbon cycling and contribute significantly to global carbon budgets (Cole et al., 2007; Laruelle et al., 2018; Regnier et al., 2013). Shelf seas are particularly susceptible to ongoing climate change due to their strong natural variability driven by complex physical and biogeochemical processes. These systems already fluctuate considerably today, and additional climate driven changes may enhance this variability or lead to unexpected responses. This interplay makes it challenging to identify long-term climate signals and complicates future predictions. Such complexity is particularly evident in the marine carbonate system (including total alkalinity (TA), dissolved inorganic carbon (DIC), pH and partial pressure of CO<sub>2</sub> (fCO<sub>2</sub>)) which display considerable spatial and temporal variation (Aufdenkampe et al., 2011; Waldbusser and Salisbury, 2014). Additional complexity arises from regional factors such as freshwater inputs from rivers and groundwater, organic matter production and decomposition, and tidal mixing (Cai, 2011b; Da et al., 2024; Salisbury et al., 2008; Song et al., 2020; Wang and Cai, 2025). Estimating the contribution of these different processes and understanding their influence on the carbonate system in coastal environments poses significant challenges compared to open ocean but is of crucial importance to predict future changes in the carbon cycle in shelf seas, regional seas, and the global ocean.

While seasonal and interannual variations in coastal carbonate systems are relatively well characterised, diel variability is often neglected, as it requires high-frequency measurements on a timescale not possible through traditional approaches. Torres et al. (2021) argued that pCO<sub>2</sub> diel cycles observed in the open ocean are predominantly controlled by thermal mechanisms, especially in low productivity waters, consistent with earlier observations by Bates et al. (1998), while in coastal oceans, these cycles are primarily governed by DIC variations and attenuated by temperature and TA (Honkanen et al., 2021; Torres et al., 2021). This underscores the need to accurately quantify the variability of the carbonate system as well as to understand the mechanisms that drive these changes. Traditional discrete water sample collection and laboratory analysis are insufficient to capture diel variability, nor do they allow the contribution of the different processes to the variation of the carbonate system. Higher-resolution and high-frequency measurements are imperative to estimate the contributions of these various processes and comprehensively assess the carbonate system dynamics in coastal regions.

In the context of these intricate coastal processes, the North Sea stands out as a biogeochemically important shelf sea. It has been the subject of extensive studies, encompassing its physical, biological and chemical conditions (Bozec et al., 2006; Thomas et al., 2005b). The Wadden Sea, situated along the North Sea coast of the Netherlands, Germany and Denmark, is a shallow intertidal sea characterised by semidiurnal tides and high biological productivity and biogeochemical cycling (Cadée and Hegeman, 2002; Christianen et al., 2017; Grunwald et al., 2010; Loebl et al., 2007; Van Beusekom et al., 2019).

90

Because the area is strongly controlled by semi-diurnal tidal cycles, exchange between the North Sea and the Wadden Sea plays an important role (Burchard et al., 2008; Elias et al., 2012; Hofmeister et al., 2017; Postma, 1954, 1961; Schwichtenberg et al., 2020a; Thomas et al., 2009; Voynova et al., 2019). The semi-diurnal tides consist of two high and two low tides per day and result in alternating shifts in water mass characteristics between the North Sea and Wadden Sea. Tide heights range from 1.5 m in the western and northern part and up to 4 m in the central part (van Beusekom et al., 2019; Schwichtenberg et al., 2020b), creating a large intertidal zone sheltered by barrier islands. In between these barrier islands, various channels facilitate exchange of water and biogeochemical constituents between the North Sea and the Wadden Sea. Several rivers, including the Elbe, Weser, Ems and IJssel, discharge into the Wadden Sea, and have historically contributed an annual average freshwater influx of approximately 60 km³ (van Beusekom et al., 2001). Moreover, driven by a northward circulation, the Rhine River plume flows along the Dutch coast, eventually reaching the Marsdiep within a month through tidal inlets east of Texel (Van Aken, 2008b). However, recent observations indicate a reduction in river flow in the Wadden Sea due to recurring extreme droughts in northwestern Europe in recent years (Philippart et al., 2024). Beyond their role as freshwater sources, these rivers are also major nutrient and alkalinity sources, resulting in a highly eutrophicated basin with pronounced salinity and nutrient gradients (Van Beusekom et al., 2001; Postma, 1981; Van Beusekom et al., 2012).

The Marsdiep channel is the largest tidal inlet in the Dutch Wadden Sea, facilitating water exchange between North Sea and western Wadden Sea (Ridderinkhof et al., 1990). With depths exceeding 40 m, the channel experiences strong tidal currents and wind-driven mixing that maintain a well-mixed water column throughout most of the tidal cycle (Buijsman and Ridderinkhof, 2008; Otto et al., 1990). This tidal exchange exerts a strong influence on carbon cycling in the southern North Sea (Brasse et al., 1999; Reimer et al., 1999) and at the same time supplies saline water to the Wadden Sea. This incoming water is a mixture of North Sea water (characterised by an approximate salinity of 35, (van Leeuwen et al., 2015)) and Rhine plume water (with salinities within a range of 30-32). The water flow reverses direction with the tide, resulting in alternating dominance of North Sea and Wadden Sea waters throughout a tidal cycle. This highly dynamic interface provides a promising location for investigating the dynamics of the marine carbonate system of both the North Sea and the western Wadden Sea and their influence on each other.

Here, we aim to determine the temporal variability of the marine carbonate system from diel to seasonal timescales at the interface between the North Sea and Wadden Sea and identify the processes involved. We use high-resolution (10 minutes) sensor data from a jetty in the Marsdiep channel, which are calibrated with measurements of discrete water samples. We collected a full seasonal cycle at a monitoring platform in the Marsdiep channel to capture temporal variation of the inorganic carbon system and other supporting parameters to characterize the temporal variation in the carbonate system parameters from diel to seasonal scales. Continuous pH data, combined with TA predicted from a multilinear regression model, enabled DIC and pCO<sub>2</sub> to be calculated which allowed us to identify and determine the processes driving the pH variability in the Wadden Sea.

### 2 Materials and Methods

### 2.1 The NIOZ jetty

We deployed sensors and collected seawater samples from the NIOZ (Royal Netherlands Institute for Sea Research) jetty monitoring platform, located at the edge of the Marsdiep (53.002 °N, 4.789 °E). The platform extends 45 m into the Marsdiep channel and has a general water depth of 3 m below N.A.P. (Amsterdam Ordnance Datum) at the monitoring platform (Fig. 1).

**Figure 1. Bathymetry map of the Wadden Sea-North Sea region.** The map on the right (b) is the zoomed-in view of the blue square indicated on the left map (a). The red cross indicates the location of the NIOZ jetty. Bathymetry data from EMODnet Digital Bathymetry (DTM 2024). Coastline data from Natural Earth.

## 2.2 Sensor measurements

The monitoring platform was equipped with a YSI EXO2 Multiparameter Sonde. Continuous temperature and practical salinity data were obtained from an EXO conductivity and temperature sensor. Salinity was measured with an accuracy and resolution of  $\pm$  1% and 0.01 respectively while temperature had an accuracy of  $\pm$  0.2 °C and a resolution of 0.001 °C, according to the

manufacture's technical specifications. In order to calibrate the salinity sensor data, discrete salinity samples were collected and analysed using a salinometer (Osil Autosal). From the start of February 2022 to the end of January 2023, 49 calibration samples were collected, with higher frequency during the growth season. During analysis, the instrument was calibrated with certified salinity standards (IAPSO standard seawater) at the beginning and at the end of each batch. The multiparameter sonde was also equipped with an EXO pH smart sensor. For a detailed discussion of the precision and accuracy of the pH measurements, see Section 3.1. Water level was also measured at the jetty station using a vegapulse 61 silo-radar level sensor.

Continuous measurements were collected by the sensors every 10 seconds; however, we used the data recorded every 10 minutes. This interval aligned with an automatic cleaning mechanism (rotating brush) designed to reduce biofouling of the sensor surfaces. By exclusively using data collected directly after cleaning, we ensured that the recorded signal accurately reflects the water signal free from interferences from biofouling. To maintain complete and constant immersion regardless of tide levels, the sensors were placed below the water surface at 1.5 m below N.A.P.

## 2.3 Discrete sampling and analysis

Discrete sampling for pH, TA and DIC was conducted between February 2022 and January 2023, covering approximately one annual cycle with only a 21-day gap. Water samples were collected from the surface using a pre-cleaned bucket deployed from the jetty platform. The effective mixing of the water column due to wind and tidal currents (Buijsman and Ridderinkhof, 2008; Postma, 1954), suggests that surface water samples are generally representative of the entire water column.

Samples were collected in 250 ml or 500 ml borosilicate glass bottles following the standard protocol for ocean carbonate system measurements (Dickson et al., 2007). During filling process, the bottles were filled slowly to avoid bubble formation and minimize gas exchange. After sample collection, a headspace equivalent to 1% of the total bottle volume was created to allow for water expansion and a mercury chloride (HgCl<sub>2</sub>) saturated aqueous solution was added to each sample (100 and 200 µl for the 250 and 500 ml bottles respectively). The samples were sealed with greased (Apiezon L) ground stoppers, secured with tape to ensure an airtight seal and prevent gas exchange, then stored in the dark at 15°C until analysis.

Measurements for TA and DIC were performed with a VINDTA 3C (Versatile INstrument for the Determination of Total inorganic carbon and titration alkalinity; Marianda, Germany) and following the standard procedures outlined by Dickson et al. (2007), as follows. DIC was determined using coulometric titration. A subsample volume of approximately 20 ml was extracted and purged of CO<sub>2</sub> by addition of 8.5% phosphoric acid (H<sub>3</sub>PO<sub>4</sub>). Subsequently, the CO<sub>2</sub> was carried through a stream of nitrogen gas through a condenser to remove any water from the gas flow, and on into a coulometric titration cell, where CO<sub>2</sub> reacts with the cathode solution forming an acid that is subsequentially titrated coulometrically. TA was measured by potentiometric titration of approximately 100 ml subsamples with 0.1 M hydrochloric acid, made up to an ionic strength similar to seawater with 0.6 M NaCl (Dickson et al., 2007). During titration, the electromotive force (EMF) resulting from acid additions was measured by a pH electrode. The TA was calculated from the titration data by the least-squares fitting in Calkulate v23.5 (Humphreys and Matthews, 2023).

Certified reference material (CRM batches 186, 195 and 205), supplied by Andrew G. Dickson from the Scripps Institution of Oceanography (USA), was used to calibrate the measurements of both TA and DIC. Multiple analyses of the CRM measurements over the full duration of the study showed a standard deviation of 0.8 and 0.9 μmol kg<sup>-1</sup> for both TA and DIC respectively. TA and DIC analyses were performed with a 1σ precision of 1.7 and 2.0 and μmol kg<sup>-1</sup> respectively.

The pH samples were measured by spectrophotometry following the protocol described by Clayton and Byrne (1993) and Liu et al. (2011). The method is based on the colour changes of the pH sensitive dye, purified meta-cresol purple, when added to a seawater sample. The analysis of the pH samples was conducted at 25 °C. Absorbances were measured both with and without the manually added dye, and following the best-practice recommendations (Dickson et al., 2007). The pH results were adjusted to the in-situ temperature using PyCO2SYS v1.8.2 (Humphreys et al., 2023). Tris buffers, provided by Andrew G. Dickson, Scripps Institution of Oceanography (USA), were used as quality control for pH measurements to ensure the proper functioning of the spectrophotometer. These buffers are used to confirm the accuracy and the repeatability of the instrument within a day of analysis. Based on duplicate samples,  $1\sigma$  precision for pH was 0.004 (n = 22) and the TRIS measurements had a standard deviation of 0.003 for TRIS batches T-34 and T-39.

## 2.4 pH sensor calibration and uncertainty calculations

### 2.4.1 pH sensor calibration

The pH sensor data, given as electromotive force (EMF) in mV, were converted into pH using the Nernst equation (Eq. (1)), which relates pH to temperature and voltage. For each discrete pH measurement, the reference potential voltage (EMF<sup>0</sup>) was calculated using the Nernst equation and the corresponding sensor EMF values. Afterwards, a piecewise cubic Hermite interpolating polynomial (PCHIP) was applied to obtain continuous EMF<sup>0</sup> values across the entire dataset. Using the interpolated values, along with the sensor's EMF values and the temperature data, the EMF values were then converted into pH using the conversion function in Calkulate v23.5 (Humphreys and Matthews, 2023). This calibration process ensured accurate calibration of the sensor's values with the pH measurement obtained through spectrophotometry throughout the deployment (Fig. 2).

$$EMF = EMF^{0} + (RT/F) * ln([H^{+}])$$
(1)

where R is the ideal gas constant (8.3145 J K<sup>-1</sup> mol<sup>-1</sup>), T is temperature (in K), and F is the Faraday constant (96.485 kC mol<sup>-1</sup>).

Figure 2. pH sensor correction using discrete pH samples. Raw uncorrected pH values (dark blue), corrected pH sensor data (dark grey), and discrete pH samples (dark circles).

#### 2.4.2 Uncertainty calculation

Uncertainty in the pH time-series measurements was estimated by adapting principles from the kriging method, a geostatistical method generally used for spatial interpolations. In the kriging method, the spatial correlation between points is analysed to predict values at unmeasured locations based on observed data at nearby locations. The spatial relationship between the observations is calculated using a semivariogram that describes how the variance between data points changes as a function of distance. By modelling these relationships, kriging can interpolate the unknown values and estimate their associated uncertainty. Here, we adapt the kriging concept to a temporal context in order to quantify the uncertainty related to the pH measurements by examining the temporal differences between all possible pairs of measurements. This method provides a robust way for assessing uncertainty in the time-series that accounts for the time since or until the nearest calibration point, helping to understand how measurement uncertainty evolves over time.

This choice of interpolation and the modified approach we employed is particularly well-suited for this study due to the challenges presented by the pH electrode sensor used to collect pH data, which is prone to drift. Furthermore, the discrete calibration samples were unevenly distributed in time. The semivariogram approach to estimating uncertainty incorporates the effects of both of these factors.

First, the offsets between the uncalibrated continuous pH sensor data and the discrete calibration pH data were calculated. Then, the absolute difference between each possible pair of offsets was computed, along with their corresponding difference in time (Fig. 3 & Fig. S1). Next, these temporal differences were binned into specific time intervals. The bin width

varied over time, starting with narrow intervals (containing between 120 to 40 values per bin) at the beginning and expanding over time where there were fewer pairs of observations (bin containing less than 40 to 1 value per bin). Within each bin, the mean of the offset values was calculated. The absolute values of the mean offsets were used to determine the standard deviations within each bin and represent how quickly the pH offset changed through time.

To model the relationship between time and uncertainty, we used a curve fitting approach (Eq. (2)) to capture the non-linear variation in uncertainty over time (Fig. 4-d):

$$\sigma(pH) = a * \sqrt{\Delta t} + b + c * \Delta t \tag{2}$$

where  $\sigma(pH)$  is the uncertainty as a function of time difference  $(\Delta t)$ , a, b, and c are least-squares fitted parameters.

Equation (2) was then used to calculate the uncertainty at each sensor measurement point while accounting, for the time between the current measurement and the previous calibration point  $(t_0)$  and the time remaining until the next one  $(t_l)$ . The uncertainties calculated for those intervals are expressed as  $\sigma_0$  and  $\sigma_l$  respectively. By weighting these uncertainties  $(\sigma_0$  and  $\sigma_l)$  and their corresponding time factors  $(t_0$  and  $t_l)$ , the uncertainty at each measurement point is expressed using the following equation:

$$uncertainty = \sqrt{t_0 \sigma_0^2 + t_1 \sigma_1^2}$$
 (3)

The uncertainty is the result of the combined uncertainties  $\sigma_0$  and  $\sigma_I$  through a weighted sum in which the weights represent  $t_0$  and  $t_I$ .

**Figure 3. Semivariogram of the temporal variability of absolute pH offset differences.** Light grey points in the background represent all possible pairs of pH offset differences, while blue points indicate the mean values calculated using moving bins.

The red line represents the fitted model. The analysis is limited to a 100-day period. The full semivariogram, including extended time intervals, is provided in the Supplementary Information (Fig. S1).

## 2.5 Determination of Total Alkalinity and Dissolved Inorganic Carbon

Based on the discrete TA measurements and the continuous salinity and water level data, a multilinear regression (MLR) model was fitted (Eq. (4)) to predict TA continuously covering the entire timeframe from February 2022 to January 2023.

$$TA_{pred} = aS + b + c\sin(2\pi(t - \varphi)/365) + d\cos(2\pi(t - \varphi)/365) + fh + g\Delta h$$
 (4)

Where TA<sub>pred</sub> is the predicted alkalinity ( $\mu$ mol kg<sup>-1</sup>), S is salinity, t is time (in days),  $\varphi$  is the phase shift of the seasonal cycle, h is the water level (cm), and  $\Delta h$  is the change in water level (cm) over the 10-minutes interval between successive measurements. Model coefficients, a through g, were optimised by least squares fitting. We note: the slope (a) = -7.084  $\mu$ mol kg<sup>-1</sup>, the intercept (b) = 2613.754  $\mu$ mol kg<sup>-1</sup>, coefficient for sine term (c) = 47.424  $\mu$ mol kg<sup>-1</sup>, coefficient for cosine term (d) = 14.630  $\mu$ mol kg<sup>-1</sup>, coefficient for water level (f) = -0.029  $\mu$ mol kg<sup>-1</sup> cm<sup>-1</sup> and coefficient for water level difference (g) = 2.104  $\mu$ mol kg<sup>-1</sup> cm<sup>-1</sup>.

The equation was based on combining a linear component linking TA to salinity, with a sinusoidal and cosine functions with a wavelength of a year to capture seasonal change. Additionally, the model also accounted for water level and its difference over time to account for hydrodynamic effects such as tidal mixing and tidal flushing. The coefficients for these variables (salinity, seasonal factors, water level and water level difference) are all adjusted for a best fit with the observed data.

Using these newly calculated  $TA_{pred}$  values and the pH sensor data, DIC and seawater pCO<sub>2</sub> (pCO<sub>2sw</sub>) were calculated at 10-minute resolution with PyCO2SYS v1.8.2 (Humphreys et al., 2023), with the carbonic acid dissociation constants of Lueker et al. (2000).

## 265 2.6 CO<sub>2</sub> sea-air fluxes

The sea-air flux of CO<sub>2</sub> (FCO<sub>2</sub>, g-C m<sup>-2</sup> day<sup>-1</sup>) was calculated as shown in equation (5). A positive FCO<sub>2</sub> indicates a release of CO<sub>2</sub> from the ocean to the atmosphere, while a negative FCO<sub>2</sub> indicates an uptake of CO<sub>2</sub> by the ocean.

$$FCO_2 = k_w * k_0 * (pCO_{2sw} - pCO_{2air})$$
 (5)

Where  $k_w$  is the gas transfer velocity (cm h<sup>-1</sup>),  $k_\theta$  is the solubility of CO<sub>2</sub> in water (mol L<sup>-1</sup> atm<sup>-1</sup>), calculated following Weiss (1974),  $p\text{CO}_{2sw}$  and  $p\text{CO}_{2air}$  represent the partial pressure of CO<sub>2</sub> in seawater and the atmosphere, respectively, both expressed in  $\mu$ atm. The  $p\text{CO}_{2sw}$  was calculated from pH and TA while the  $p\text{CO}_{2air}$  was derived by converting the CO<sub>2</sub> mole fraction (xCO<sub>2</sub>) to  $p\text{CO}_{2air}$ , using PyCO2sys v1.8.2 (Humphreys et al., 2023).

The transfer gas velocity was calculated using the formulation of Nightingale et al. (2000), which was established in the North Sea, with a Schmidt number (*Sc*) normalised to a *Sc* of 600:

$$k_w = (0.333 * U + 0.222 * U^2) * (600/S_c)^{0.5}$$
(6)

The *U* is the wind speed (in m s<sup>-1</sup>). Hourly wind data was acquired from the Royal Netherlands Meteorological Institute (KNMI) for the Texelhors wind platform and interpolated to a 10-min interval, in order to match the resolution of the collected dataset.

The  $K_{\theta}$ , Sc and  $K_{w}$  were calculated using in-situ temperature and salinity measurements collected during the study period. All calculations were computed using pySeaFlux v2.0.0 (Gregor and Humphreys, 2021).

#### 285 **2.7 Wasserstein Distance**

DIC was decomposed into components driven by biological and hydrological processes: DIC<sub>bio</sub> and DIC<sub>hydro</sub>. By examining the relative contributions of each component of DIC, we can then understand how seasonal changes in these processes impact DIC, and consequently, pH.

DIC<sub>bio</sub> and DIC<sub>hydro</sub> were calculated using PyCO2SYS v1.8.2 (Humphreys et al., 2023). For DIC<sub>bio</sub>, changes in DIC were considered to be driven by pH, while for DIC<sub>hydro</sub>, variations in TA, salinity and temperature account for changes in DIC. DIC<sub>bio</sub> was calculated using the daily averages for TA, salinity and temperature but with varying pH. In contrast, DIC<sub>hydro</sub>, was calculated using the daily averages of pH, with varying TA, salinity and temperature (Eqs. 7 and 8). The daily average TA used for the calculation of DIC<sub>bio</sub> was recalculated (as described in Section 2.5) with daily averages of salinity and water level.

DIC<sub>bio</sub> = PyCO2SYS(pH<sub>obs</sub>, TA<sub>avg</sub>, 
$$S_{avg}$$
,  $T_{avg}$ ) (7)

$$DIC_{hydro} = PyCO2SYS(pH_{avg}, TA_{obs}, S_{obs}, T_{obs})$$
(8)

The contributions of biological and hydrological processes to the observed DIC were estimated using the Wasserstein distance, a method based on optimal transport theory. Conceptually, optimal transport theory is pictured transporting piles of sand from one location to another without any gain or loss of material. The goal is to identify the most efficient way of moving the sand, by minimising the amount of sand to be moved and the distance to be travelled. In this case, the Wasserstein distance represents the cost associated with the most efficient transport (Hyun et al., 2022; Lipp and Vermeesch, 2023; Peyré and Cuturi, 2020; Villani, 2009).

In the context of this study, the sand piles represent DIC values. The Wasserstein distance compares the predicted (DIC<sub>bio</sub> and DIC<sub>hydro</sub>) and the observed DIC, by capturing both the magnitude and timing of their differences. This is particularly important as DIC is influenced by both biological and hydrological processes over time. Standard metrics, such as the root

mean square deviation (RMSD), compare values with identical points in time, which can overestimate discrepancies when the predicted and observed values are shifted in time. The Wasserstein distance overcomes this limitation by comparing the overall patterns. This approach allows us to determine which of the two processes, the biological or hydrological, follows more closely the observed patterns of DIC.

## 3 Data quality

## 3.1 pH sensor performance

Over the 12-month of study, the pH sensor recorded a total of 53203 measurements at 10-minute intervals. Of these, 292 measurements (representing 0.55% of the entire dataset) were recorded while the sensor was removed from the water for maintenance and calibration and were therefore flagged as unusable and excluded from the dataset for all subsequent analysis.

Throughout the study period, discrete pH samples were collected directly from the jetty to calibrate the pH sensor and correct for potential drifts. The sampling frequency varied across the year: from March to July, up to 13 samples were collected per month, while no sample was collected during November. From late October through January, only two samples were collected in total. This irregular sampling frequency is taken into considerations for sensor calibration and uncertainty estimations.

Comparison between raw pH sensor data (converted from mV to pH using Eq. (1)) and the calibration samples measured by spectrophotometry showed that the pH sensor consistently read higher values than the calibration measurements (Fig. 2). The mean offset was  $0.28 \pm 0.09$  with a total range of 0.1 to 0.42, which aligns with observations reported by Johengen et al. (2015) for the same pH sensor. Additionally, these offsets showed a temperature dependency, where higher temperatures were associated with larger offsets (Pearson's r = 0.528, p 

against observed pH variations. Seasonal variation, based on monthly averages, reached  $0.255 \pm 0.077$ , while diel fluctuations averaged  $0.092 \pm 0.061$ . Both signals exceeded the maximum estimated uncertainty, indicating that the pH sensor data retained adequate resolution to capture these patterns.







# Figure 4. Seasonal variability of temperature, salinity, TA, DIC, pH, ΔfCO<sub>2</sub>, and FCO<sub>2</sub>

Discrete measurements of TA (c), pH (d) DIC (e) are represented by dark circles while continuous measurements are shown in blue. The pink shaded area around the pH sensor data (d) represents the uncertainty associated with the pH sensor measurements. Cumulative flux (g) is shown in a solid line.  $\Delta fCO_2$  (f) and  $FCO_2$  (g) are plotted with a horizontal dashed line at zero indicating the equilibrium point.

#### 3.2 MLR validation

### 3.2.1 Overall performance

The MLR performance was assessed using different statistical metrics, including the coefficient of determination  $R^2$ , the RMSD and the Nash-Sutcliff model efficiency (NSE) (Nondal et al., 2009), which assess how well model predictions match the observed data. The NSE values close to 1 indicate perfect model performance, values near 0 suggest the model performs similarly to using mean of observations and negative values to indicate that the model performs worse than the mean.

The comparison between  $TA_{pred}$  and measured TA showed that the MLR (Eq. (4)) successfully captured the main observed pattern and variability (Fig. 4-c). The RMSD between the measured TA and  $TA_{pred}$  was 17.5  $\mu$ mol kg<sup>-1</sup> with an R<sup>2</sup> of 0.82. This RMSD is considerably smaller than both the observed seasonal TA range (231.7  $\pm$  12.7  $\mu$ mol kg<sup>-1</sup>) and the average daily range (46.4  $\pm$  13.1  $\mu$ mol kg<sup>-1</sup>), representing 7.6  $\pm$  0.7 % of the seasonal variation and 37.7  $\pm$  11.0 % of the daily variation.

The time series (Fig. 4-c) and the high-resolution data (Fig. 5-a) confirm that the model successfully tracked the seasonal and daily TA cycles, reproducing the magnitude and timing of variations accurately enough for our purposes. Across the 59 TA observations the model showed a strong linear relationship, however, tended to underestimate TA in the lower and higher ranges ( $<2375 \,\mu$ mol kg<sup>-1</sup>,  $>2450 \,\mu$ mol kg<sup>-1</sup>) and overestimated it in the mid-range ( $\sim2375-2450 \,\mu$ mol kg<sup>-1</sup>) (Fig. S2-a). Additionally, the MLR predictions showed some oscillation patterns that are not present in the discrete TA observations, which could be attributed to the annual harmonic terms that capture the seasonal variation across the study period. Nevertheless, the model demonstrates strong performance for seasonal scale predictions and reasonable ability to capture daily fluctuations.

To further validate the TA prediction model, we evaluated its performance on a shorter time scale where high-resolution sampling was conducted from March 29<sup>th</sup> to April 1<sup>st</sup>, during which 12 samples of TA, DIC and pH were collected (Fig. 5). These samples were not used to develop nor train the MLR, ensuring unbiased validation. Despite the reduced temporal scale, the RMSD remained smaller, representing  $26.6 \pm 3.7$  % of the daily ranges ( $65.8 \pm 3.2 \,\mu mol \, kg^{-1}$ ), confirming the model's performance on both short and long-term scales.

Correlation analysis between the MLR components and TA revealed varying relationships (Table 1). Seasonal variability (sine component) exhibited the strongest correlation with TA, followed by temporal trends and salinity. Water level difference showed no direct significant correlation with TA. However, including it in the MLR significantly improved the MLR performance with R<sup>2</sup> increasing from 0.76 to 0.82, RMSD decreasing from 20.3 to 17.5 µmol kg<sup>-1</sup> and the NSE increasing






from  $0.76 \pm 0.008$  to  $0.82 \pm 0.006$ . Model performance increased by  $8.2 \pm 1.35$  % and a reduced uncertainty of  $2.8 \pm 2.5$  µmol 405 kg<sup>-1</sup>. While this improvement suggests that water level captured a real tidal influence on TA, its inclusion in the MLR may also create TA-tidal pattern similarities.

The MLR model was also assessed for potential biases. Results showed no significant correlations between MLR components and alkalinity residuals (TA<sub>res</sub>) (Table 2), which confirms that the model accurately captures the main alkalinity variation without systematic bias. The TA<sub>res</sub> times series (Fig. S2-b) confirms no temporal bias, with residual variability reflecting the irregular sampling frequency.

The monthly model performance revealed varying RMSD closely linked to sampling frequency (Fig. S2-c). During high sampling periods (March-July, n=7-13), RMSD estimates remained relatively consistent, ranging between 15.3 and 21.3 µmol kg<sup>-1</sup>. In contrast, months with limited sampling (n=1-2), showed unreliable RMSD estimates, as insufficient observations cannot accurately capture the real monthly variability needed by the model to predict TA. These results reflect a sampling limitation rather than model deficiency.

The overall RMSD estimates (17.5 µmol kg<sup>-1</sup>) compare positively with other coastal MLR studies. While global ocean TA prediction typically achieves uncertainties lower than 10 μmol kg<sup>-1</sup> (Carter et al., 2016; Lee et al., 2006; McNeil et al., 2007), MLR performances vary considerably with system complexity and heterogeneity. Regional-scale coastal studies achieve lower uncertainties, where for instance Gemayel et al. (2015) reported an uncertainty of 10.6 µmol kg<sup>-1</sup> across the Mediterranean Sea, while Alin et al. (2012) reported an uncertainty of 6.4 µmol kg<sup>-1</sup> along the South Californian coast. Complex coastal systems exhibit substantially higher uncertainties, as indicated by Rheuban et al. (2021) with uncertainties ranging from 48 to 121.1 μmol kg<sup>-1</sup> across various estuarine systems along the northeast US coast. Similarly, Fassbender et al. (2017) achieved identical performances as in this study with an uncertainty of 17 µmol kg<sup>-1</sup> in Washington state nearshore and estuarine environments. Comparing the high RMSD observed in this study with the RMSD estimated for open ocean or regional coastal system reflects the biogeochemical complexity of the Wadden Sea. TA in these systems is primarily controlled by conservative mixing processes and presents a strong correlation with salinity. However, in the Wadden Sea various biogeochemical processes, such as denitrification and sulfate reduction in anoxic sediments contribute to TA generation (Thomas et al., 2009). Moreover, organic compounds in seawater can also contribute to TA (Goldman and Brewer, 1980), and this is known as organic alkalinity (OrgAlk). In order to assess the contribution of OrgAlk to the MLR prediction errors, OrgAlk was estimated from subtracting calculated TA (TAcalc) from measured TA, with TAcalc calculated from DIC and pH using PyCO2SYS v1.8.2 (Humphreys et al., 2023). OrgAlk showed weak correlation with salinity and TA<sub>res</sub> (Fig. S3; R<sup>2</sup> = 0.292 and  $R^2 = 0.061$ , respectively), indicating that OrgAlk does not explain the MLR prediction errors. Therefore, likely other processes, such as denitrification and sulfate reduction in the Wadden Sea, are affecting TA variability, which are not (fully) captured by the MLR model.

Overall, based on the conducted statistical tests and the degree of complexity of the MLR, which included tidal and temporal influences, the overall MLR is adequate for the application in this study for seasonal and daily TA variability.




**Table1. MLR component correlation with alkalinity.** Where r is the coefficient of correlation,  $R^2$  is the coefficient of determination and p-value is the statistical significance (p < 0.05 means statistically significant).

| MLR component   | r      | $\mathbb{R}^2$ | p-value |
|-----------------|--------|----------------|---------|
| $\sin(2\pi t)$  | 0.824  | 0.679          | < 0.001 |
| Time (t)        | -0.703 | 0.494          | < 0.001 |
| Salinity (S)    | -0.529 | 0.280          | < 0.001 |
| cos(2πt)        | 0.318  | 0.101          | < 0.05  |
| Water Level (h) | -0.262 | 0.069          | <0.05   |

-0.135

Table 2. MLR component correlation with alkalinity residuals. Where r is the coefficient of correlation,  $R^2$  is the coefficient of determination and p-value is the statistical significance (p < 0.05 means statistically significant).

0.309

0.018

| MLR component               | r      | $\mathbb{R}^2$ | p-value |
|-----------------------------|--------|----------------|---------|
| $\sin(2\pi t)$              | 0.000  | 0.000          | 1.000   |
| Time (t)                    | 0.050  | 0.000          | 0.709   |
| Salinity (S)                | -0.000 | 0.000          | 0.999   |
| cos(2πt)                    | -0.000 | 0.000          | 0.998   |
| Water Level (h)             | 0.006  | 0.000          | 0.996   |
| Water Level Difference (Δh) | -0.127 | 0.016          | 0.336   |

# 3.2.2 DIC calculations from TA<sub>pred</sub>

Water Level Difference ( $\Delta h$ )

DIC was calculated from  $TA_{pred}$  and the calibrated pH sensor data. The calculated DIC, DIC<sub>calc</sub>, closely aligns with measured DIC through the year, effectively capturing the patterns observed in the measured DIC (Fig. 4-e). This similarity could reflect a calculation dependency, as DIC<sub>calc</sub> was estimated form pH and  $TA_{pred}$ . However, the measured DIC values exhibited comparable values and patterns, independently from  $TA_{pred}$ , confirming then that DIC<sub>calc</sub> accurately capture real DIC variability, within calculation uncertainties.

The RMSD between the measured DIC and DIC<sub>calc</sub> was 19.8  $\mu$ mol kg<sup>-1</sup>, with an R<sup>2</sup> of 0.87. Considering the observed seasonal variation in DIC of 419.3  $\pm$  4.9  $\mu$ mol kg<sup>-1</sup>, the model uncertainty accounted for only 4.7  $\pm$  0.5 % of the seasonal range and 27.1  $\pm$  11.0 % of the average daily variation (73.1  $\pm$  28.8  $\mu$ mol kg<sup>-1</sup>).




The high-resolution validation period, as described above, was also used to assess the DIC<sub>calc</sub> (Fig. 5-b). During this period, the RMSD remained small compared to the observed daily ranges (75.1  $\pm$  1.3  $\mu$ mol kg<sup>-1</sup>), with model uncertainty representing 26.4  $\pm$  4.0 % of the daily variation, confirming the model's performance on predicting TA<sub>pred</sub> and its reliability in using the predicted values to calculate DIC<sub>calc</sub>. However, the DIC<sub>calc</sub> slightly overestimated the measured values, especially during the first day of this high-resolution sampling period, possibly due to an underestimation of the hydrodynamic effects in TA<sub>pred</sub>.

Additionally, to assess the sensitivity of DIC<sub>calc</sub> to pH calibration, the pH sensor data was recalibrated by progressively omitting measured pH values, reducing the number of measured pH values used during calibration from 12 calibration samples up to scenarios where only one pH calibration sample remained. Those changes were applied only during the high-resolution sampling period. For each scenario, the recalibrated pH, combined with TA<sub>pred</sub>, was then used to recalculate DIC<sub>calc</sub>, and then compared to those calculated from the fully calibrated pH sensor data. The results showed minimal variation in DIC<sub>calc</sub> across all calibration scenarios. The single-point calibrated data DIC<sub>calc</sub> and fully calibrated DIC<sub>calc</sub> showed substantial overlap (Fig. S4-a). As expected, the pH values changed when using fewer calibration samples compared to the full calibration dataset (Fig. S4-b). Statistical analysis revealed no significant differences between the DIC<sub>calc</sub> with fully pH calibrated data and DIC<sub>calc</sub> under different calibration scenarios (p-value > 0.05). This suggests that variation in TA had a greater influence on DIC than changes in pH.

Figure 5: high-resolution data of TA, DIC and pH from March 29th to April 1st, 2022.

The dark circles represent collected TA, DIC and pH samples. Continuous lines depict: (a) TA<sub>pred</sub>, (b) DIC<sub>calc</sub> and (c) calibrated pH sensor data. The red circle in panel (c) highlights the single pH sample from this period that was used for pH sensor calibration. The data were collected over a 72-hour period. Vertical lines separate the days.

## 4 Results and discussion

## 4.1 Seasonal cycle

## 4.1.1 Seasonal hydrological influence

The water temperature varied from a minimum of around  $4 \pm 0.2$  °C in December to  $23 \pm 0.2$  °C in August (Fig. 4-a). Salinity followed a typical pattern (Ridderinkhof et al., 2002) with a less clear cycle, being lower and more variable in the winter and higher and more consistent in the summer (Fig. 4-b). Salinity is generally lower in winter because precipitation, especially







over the land, is higher, which increases the freshwater flow from the IJsselmeer, other rivers flowing directly into the Wadden Sea, and the Rhine plume entering through the Marsdiep tidal inlet. The higher variability in salinity arises because the greater precipitation is strongly linked to ephemeral stormy weather (Van Aken, 2008b). However, increased salinities during summer periods, especially during the recent years with increased extreme droughts, are likely due to reduced or completely stopped discharge of freshwater from the IJsselmeer into the Wadden Sea. This water management approach is implemented in order to preserve freshwater for agricultural use and consumption (Philippart et al., 2024).

TA presented to first order an opposite pattern to salinity, with higher values in the winter/spring ( $\sim$ 2500  $\pm$  17.5  $\mu$ mol kg<sup>-1</sup>) and lower in the summer/autumn ( $\sim$ 2350  $\pm$  17.5  $\mu$ mol kg<sup>-1</sup>), but its variability was more consistent throughout the year (Fig. 4-c). The opposite patterns of TA and salinity arose because the low-salinity endmember water mass generally had higher TA than the high-salinity endmember (Fig. 6). However, this TA-salinity relationship varied seasonally. From the regressions between TA and salinity for different months, both TA and salinity of the low endmember displayed large range of variation (163.6  $\pm$  65.2  $\mu$ mol kg<sup>-1</sup> and 5.1  $\pm$  2.1 respectively), while the high endmember exhibited more variable TA but relatively stable salinities (115.0  $\pm$  43.6  $\mu$ mol kg<sup>-1</sup> and 3.0  $\pm$  1.0 respectively). This suggests different Wadden Sea water masses becoming more or less dominant at different times (Hoppema, 1990, 1993).

The lack of freshwater end-member data prevents us here from establishing a reliable relationship between TA, DIC and salinity. As observed in Fig. 6, different TA values were recorded for the same salinity values, indicating the influence of mixing between multiple endmembers. This variability is the result of the dominance or lack of dominance of different water masses in the Wadden Sea depending on the hydrodynamic conditions. The western Wadden Sea in particular exhibits a strong gradient and highly variable hydrographic characteristics, both on tidal and longer time scales. This variability is due to multiple factors including, tidal exchange, variable freshwater discharges and changing Dutch North Sea coastal waters properties (Postma, 1954). Therefore, understanding the TA-S relationship is essential for estimating the characteristics of the different water masses which interact based on hydrographic conditions (the Wadden Sea endmember, the IJsselmeer endmember and the North Sea endmember) and calculating the influence of biogeochemical processes on the carbonate system. Without direct measurement of the freshwater endmember, the TA and DIC seasonal variations can only provide indirect insights into potential sources and sinks of TA, independently of DIC fluctuations. Consequently, understanding the interplay between tidal cycles, freshwater discharges and biogeochemical processes is essential, yet challenging, to fully characterise and understand the carbonate system in the Wadden Sea. Further research incorporating freshwater end members, along other measurements, will be essential in order to refine our understanding of these interactions.

## 555 Figure 6. Regression between salinity and alkalinity in the Marsdiep Channel.

Data points are coloured by month. Regression lines are fitted for grouped months (December-March, July-August, September-October) and individual months.

## 4.1.2 Net community production and respiration



pH exhibited strong seasonal variations, with a maximum of  $8.35 \pm 0.04$  during spring and a minimum of  $7.75 \pm 0.02$  during summer (Fig. 4-d). The high pH observed in spring was likely due to a phytoplankton bloom. Through March, DIC calc dropped from 2426.8 to 2180.7  $\pm$  19.8  $\mu$ mol kg<sup>-1</sup>, presumably due to conversion of DIC into organic matter, increasing pH by  $\sim$ 0.3  $\pm$  0.06 (Soetaert et al., 2007). This inverse relationship between pH and DIC during short-term (days to weeks) excursions continued through the year, although on longer time scales (months), the pattern of DIC was more like that of TA<sub>pred</sub>, with a minimum DIC of 2007  $\pm$  19.8  $\mu$ mol kg<sup>-1</sup> during summer and a maximum of 2426.8  $\pm$  19.8  $\mu$ mol kg<sup>-1</sup> in March 2022 (Fig. 4-e). The short-term excursions in pH and DIC suggest the occurrence of smaller, secondary plankton blooms later in the year, while the longer-term similarity with TA suggests a relatively constant state of (dis)equilibrium with atmospheric CO<sub>2</sub> after the spring bloom (Humphreys et al., 2018), as seen from the  $\Delta$ fCO<sub>2</sub> (Fig. 4-f).

In February, FCO<sub>2</sub> (Fig. 4-g) reached a maximum of 1.7 g-C m<sup>-2</sup> d<sup>-1</sup>, a CO<sub>2</sub> source to the atmosphere, while through March and into early April, the decline in DIC caused it to drop to a minimum of -1.4 g-C m<sup>-2</sup> d<sup>-1</sup>, a CO<sub>2</sub> sink (Thomas et al.,






2005a). The cumulative FCO<sub>2</sub> since the start of our observations on 2 February reached a maximum of 0.1 g-C m<sup>-2</sup> total CO<sub>2</sub> released to the atmosphere on 13 March, but almost as quickly returned to zero by the beginning of May.

From May until January, FCO<sub>2</sub> fluctuated moderately within a range of -0.6 to 0.5 g-C m<sup>-2</sup> d<sup>-1</sup> while the cumulative FCO<sub>2</sub> remained negative, reaching a minimum of -2.2 g-C m<sup>-2</sup> total CO<sub>2</sub> taken up from the atmosphere on 10 July. The cumulative FCO<sub>2</sub> showed periods of increase, interspersed with two short decreases presumably linked to the secondary blooms observed in July and September. Despite these fluctuations, an overall increase in cumulative FCO<sub>2</sub> from May to January illustrated the Wadden Sea's role as a source of CO<sub>2</sub> through summer, autumn and winter, except during brief bloom-related uptake periods (Clargo et al., 2015; Thomas et al., 2005a). The positive cumulative FCO<sub>2</sub> by January of 4.7 g-C m<sup>-2</sup>, despite a 21-day gap to complete a full year, indicates a net CO<sub>2</sub> release to the atmosphere and suggesting that the Wadden Sea acts as a CO<sub>2</sub> source over the entire annual cycle, consistent with previous findings (Thomas et al., 2004).

The negative correlation between TA and salinity, combined with similar TA and DIC seasonal variability, highlights an important characteristic of the Wadden Sea carbonate system. Seasonal variability in both TA and DIC was dominated by hydrological process (Section 4.1.1), with the additional biological effect on DIC difficult to discern. However, because the hydrological effects on TA and DIC were well correlated with each other, they mostly cancelled each other out in terms of their effects on pH. This results in biological processes being the primary driver of seasonal pH variability.

# 4.1.3 Other processes

Beyond net community production (NCP) and respiration, other biological processes also influence the carbonate system in the Wadden Sea. The available nutrient data that was collected during the study (Fig. S5) exhibited similar patterns and ranges to what was observed by Salt et al. (2014) in the Marsdiep channel, with high nutrient concentrations during autumn and winter and lowest concentrations during spring and summer. Salt et al. (2014) identified nitrification and ammonia uptake as key drivers of TA consumption, whereas denitrification and sulfate reduction act as drivers of TA production. Additionally, their study highlighted the role of silicate and nitrate availability on NCP during diatom blooms.

Given the similarities (ranges and patterns) in nutrients, TA and DIC distributions with the results of Salt et al. (2014), denitrification and sulfate reduction processes may be driving TA generation in the Wadden Sea as well. Thomas et al. (2009) estimated that these processes generate approximately 99 Gmol TA yr<sup>-1</sup> and 12-26 Gmol TA yr<sup>-1</sup> respectively. The importance of these sedimentary processes for the carbonate system of the Wadden Sea are well documented in many previous studies (Al-Raei et al., 2009; Brenner et al., 2016; Kieskamp et al., 1991; Norbisrath et al., 2024; Thomas et al., 2009; Voynova et al., 2019). All these processes might help explaining the observed TA and DIC distributions.

As mentioned above, both TA and DIC showed similar patterns with elevated concentrations in spring, followed by a gradual decrease through summer and autumn before increasing again in winter. The monthly variability in the  $\Delta TA/\Delta DIC$  ratio (Fig. S6) can provide additional insights on the dominant biogeochemical processes occurring. The  $\Delta TA/\Delta DIC$  ratio exhibits a clear seasonality, with maximum value in winter (0.89  $\pm$  0.01), lower values in spring and summer, and a negative ratio in September (-0.08  $\pm$  0.01). The winter higher ratios might coincide with TA being generated at higher rate than DIC







and could result from denitrification. The peaks during winter suggest that when NCP is minimal, the influence of sedimentary processes on the carbonate system variability becomes more apparent in the water column, as their signature is not as much obscured by NCP removal of DIC. However, the lower ratios during spring and summer, reflect on the dominance of NCP in removing DIC at rates faster than TA generation, resulting therefore in lower  $\Delta TA/\Delta DIC$  values. The negative ratio in September (-0.07  $\pm$  0.01) could represent a transitional period between the decline of NCP and an increase in organic matter remineralisation.

## 4.2 Diel cycles

### 4.2.1 Influence of tidal forcing

The diel cycles of TA, salinity, and water level were distinctly different from those of temperature, pH, DIC and  $\Delta f$ CO<sub>2</sub>. Monthly-averaged hourly TA anomalies with respect to the daily mean (Fig. 7 & Fig S7) varied between  $-7.2 \pm 4.3$  and  $7.0 \pm$ 4.3  $\mu$ mol kg<sup>-1</sup>. However, individual daily anomalies showed much larger variability, ranging between -43.4  $\pm$  15.0  $\mu$ mol kg<sup>-1</sup> and 44.3 ± 9.6 μmol kg<sup>-1</sup>, displaying a complex, irregular pattern without a clear day-night cycle, as the tidal cycle likely conceals any underlying biological patterns, and suggesting that tidal processes dominate over NCP and respiration in controlling daily TA variation. Changes in TA appeared to be associated with the tidal cycle, with fluctuations observed in response to both high and low tides (Fig. S8). Higher TA was observed during low tide. During high tide, North Sea water, which has lower TA than the Wadden Sea, is brought into the Wadden Sea, lowering the overall TA. During low tide, the dominance of the Wadden Sea water, which has higher TA, results in increased TA. Water level monthly anomalies (Fig. S8), which fluctuated between  $-27 \pm 39$  cm and  $29.2 \pm 45$  cm (with individual daily anomalies ranging from  $-155 \pm 55$  to  $149 \pm 55$ cm), also showed a complex and irregular diel pattern driven by the tides, and consistent with a dominant M2 tidal constituent (Ridderinkhof et al., 2002b). The Wadden Sea is characterised by asymmetrical tides, where the ebb and flood tides fluctuate in both timing and intensity (Buijsman and Ridderinkhof, 2007). This tidal asymmetry combined with the delay in TA response to changes in water level – which is typically less than 5 hours – means that the minima and maxima in water level did not correspond exactly with the those in TA, and vice versa. This further complicated the overall tidal pattern by creating uneven fluctuations in water level, making it difficult to establish regularity in the tidal cycle despite the underlying tidal control.

These patterns were further examined through tidal phase analysis (Fig. 8 & Fig S9). While several months exhibited the expected pattern of higher TA at low tide and consistent with the mixing mechanism described above, other periods displayed opposite behaviours. This variability does not follow a clear seasonal pattern, as months from different seasons exhibited similar variability, while months within the same season showed different patterns. This irregularity reflects the complexity of TA dynamics in a shallow tidal system. Tidal forcings lead to strong benthic-pelagic coupling (Huettel et al., 2003), moreover, asymmetrical tides combined with the 



Salinity monthly anomalies (Fig. 7 & Fig. S10), which varied between  $-0.7 \pm 1.2$  and  $1.0 \pm 1.9$  (with individual daily anomalies ranging from  $-5.2 \pm 0.7$  to  $5.4 \pm 1.5$ ), displayed a clearer 12-hour periodicity compared to TA and water level. The semi-diurnal tides in the Wadden Sea (Buijsman and Ridderinkhof, 2007; van der Molen et al., 2022), consisting of two low and two high tides per day, lead to alternating shifts in TA and salinity as the system transitioned from Wadden Sea to North Sea dominance. During high tide, the influx of North Sea waters introduces higher salinity water into the Wadden Sea, resulting in a more homogeneous salinity distribution. At low tide conditions, local estuarine processes and freshwater inputs create more variable salinity conditions. However, in dry summer periods or drought conditions, particularly when freshwater discharges are reduced or completely stopped (Fig. S11), the salinity gradient between North Sea and Wadden Sea can be reversed.

Salinity also exhibited seasonal variation in the intensity of its diel fluctuations. These seasonal patterns were further examined across the tidal phase (Fig. 8 & Fig. S12), confirming the varying intensity of tidal influence on salinity variation. In March, salinity anomalies were particularly pronounced, reaching their minimum values at low tides and maximum values at high tide, and indicating strong tidal control of salinity variability. From May onward, this variability declined progressively until reaching near zero in August and September. The reduced salinity variability corresponds to a decrease in freshwater 650 discharge from the Ijsselmeer and the Rhine during summer (Fig. S11), resulting in a weakened and more uniform salinity gradient between the Wadden Sea and the inflowing North Sea waters.

From October to December salinity anomalies increased again, likely driven by seasonal changes in freshwater input and mixing, during which tidal control of salinity variability strengthened again as the freshwater-seawater gradient was reestablished.



## 4.2.2 Influence of biological processes

Across the year, pH, DIC and  $\Delta f CO_2$  exhibited clearer 24-hour periodicities, particularly during the warmer months (Mayaugust). The largest monthly amplitude pH anomalies (Fig. 7 & Fig. S13), occurred in June, with a night-time minimum of - $0.03 \pm 0.03$  and a daytime maximum of  $0.03 \pm 0.04$ . In winter, variations were smaller, often near zero. The diel pH cycle displayed a characteristic day-night pattern, consistent with biologically driven CO2 dynamics associated with NCP and respiration. During the day, photosynthesis processes consume CO<sub>2</sub>, reducing the concentration of CO<sub>2</sub> and therefore increasing pH. At night, respiration and remineralisation are the dominant biological process releasing CO<sub>2</sub> back in the seawater, and lowering pH (Bates et al., 2009; Falkowski, 1994). These mechanisms have been widely documented in coastal systems (Baumann and Smith, 2018; Cai, 2011a; Gattuso et al., 1998), particularly in shallow tidal systems (Bauer et al., 2013; Baumann and Smith, 2018; Cai, 2011a).


The magnitude of diel pH fluctuations varied seasonally, with greater amplitude in spring and summer compared to winter. This was particularly evident in July and August where the daily peak of pH occurred several hours later compared to








June, and potentially reflected the effect of temperature. In wintertime, the daily variation in pH was weaker, likely due to diminished NCP and lower sea surface temperatures. In addition, mixing between the Wadden Sea and the southern North Sea likely altered pH and obscured an already reduced biological signal.

Tidal phase analysis (Fig. 8 & Fig. S14) revealed seasonal patterns in pH variability. During winter (February, November-January), pH anomalies varied minimally across tidal phases, consistent with reduced biological activity. However, from March through July, pH anomalies displayed a more complex pattern with a distinctive w-shaped pattern across the tidal cycle. This pattern developed gradually from March, suggesting a strengthening of the biological processes during this period and creating increasing biogeochemical differences between the North Sea and Wadden Sea during the high productive season. This seasonal transition was further supported by a Fourier analysis (Fig. S15), which revealed that from June to September the 24-h frequency dominated the 12-h frequency, suggesting a much stronger influence of biological processes on the daily cycle. However, during the rest of the year, particularly in winter when NCP is lower, the 24-h/12-h ratio was smaller, suggesting that the tidal forcing played a more prominent role in timing pH variability as result of the mixing between the different water masses in the Marsdiep Channel.

The diel cycle of temperature anomalies also varied seasonally (Fig. 7 & Fig. S16). Day-night temperature differences were most pronounced from March through August, reaching a maximum in May with monthly anomaly amplitudes of  $0.6 \pm 0.2$  °C (daily anomalies amplitude of  $3.7 \pm 0.4$ ). These fluctuations decreased until reaching a minimum in January with anomaly amplitudes of  $0.2 \pm 0.04$  °C.

Although temperature influences pH by altering CO<sub>2</sub> solubility, with warmer waters generally characterised with higher pH due to reduced CO<sub>2</sub> solubility (Zeebe and Wolf-Gladrow, 2001), this mechanism alone cannot explain the observed patterns. While the carbonate system equilibration timescale is on the order of minutes or faster (Mojica Prieto and Millero, 2002), the exchange of CO<sub>2</sub> between the atmosphere and sea reaches equilibrium over much longer timescales, from weeks to months (Jones et al. (2014)). Therefore, if temperature were the main driver, pH would be expected to closely follow the temperature pattern. To assess the influence of temperature, pH was recalculated using varying temperature and monthly means of TA, DIC and salinity (pH<sub>temp</sub>) using PyCO2SYS v1.8.2 (Humphreys et al., 2023) (Fig. S17). The observed pH variations appeared to be primarily driven by processes other than temperature change, as key variations often occurred independently of temperature fluctuations. Additionally, pH deviated from pH<sub>temp</sub> for most of the year, especially during the warmer months. For instance, pH often reached its daily maximum several hours before temperature, as observed in June (Fig. 7, Fig. S13 & Fig. S17). However, in the beginning of February, then December and January, pH tracked closely pH<sub>temp</sub>, indicating that temperature's influence on pH became more prominent during the colder months, during which biological activities were reduced (Fig. S17).

If temperature were the primary driver, pH should consistently rise and fall with temperature fluctuations through the year, but this was not observed during this study. This strongly suggests that biological activities played a more dominant role in influencing pH variations. The mismatch between temperature and pH peaks is more consistent with light-driven NCP, during which CO<sub>2</sub> is consumed and results in a pH increase during periods of maximum light availability rather than maximum






temperatures. This biological control may also (partly) explain the seasonal variation in pH amplitudes, with higher variability in spring and summer and coinciding with periods of increased NCP.

The diel cycle of DIC anomalies (Fig. 7 & Fig. S18) was opposite to that of pH. In winter, DIC monthly anomalies were smaller and ranged between  $-6.6 \pm 3.2$  and  $5.9 \pm 4.0$  µmol kg<sup>-1</sup> (with daily anomalies ranging from  $-43.4 \pm 12.6$  µmol kg<sup>-1</sup> to  $50.4 \pm 12.7$  µmol kg<sup>-1</sup>), consistent with weaker biological activities. Whereas, in spring and summer, these anomalies increased up to  $17.6 \pm 13.3$  and  $22.2 \pm 13.3$  µmol kg<sup>-1</sup> (with daily anomalies ranging from  $-90.5 \pm 26.8$  µmol kg<sup>-1</sup> to  $103.1 \pm 18.1$  µmol kg<sup>-1</sup>). These results suggest intensified respiration during the night and NCP during the day. During these periods, pH changes closely followed DIC fluctuations, with nighttime increases in DIC corresponding to decreases in pH, and daytime DIC decreases aligning with pH increases. For instance, in June a decrease of  $39.9 \pm 13.3$  µmol kg<sup>-1</sup> in DIC anomalies over 7 hours during the day corresponded to a pH increase of  $0.06 \pm 0.02$  during the same period, suggesting a link between DIC and pH variability that is answering to NCP and respiration. Tidal phase analysis of DIC (Fig. 8 & Fig. S19) revealed similar complexity as TA which suggests that, like TA, DIC is controlled by multiple mechanisms including water mass mixing, biological processes and potentially benthic fluxes that can mask the expected tidal signal, creating a complex DIC-tidal phase relationship.

 $\Delta$ /CO<sub>2</sub> (Fig. 7 & Fig. S20) mirrored the diel pattern of DIC.  $\Delta$ /CO<sub>2</sub> anomalies were typically negative during daytime as the sea acted as a CO<sub>2</sub> sink and shifted to positive anomalies at night, likely due to respiration. This diel  $\Delta$ /CO<sub>2</sub> cycle became more pronounced from March to October, with monthly anomalies ranging from -37 ± 49 to 43 ± 47  $\mu$ atm (with individual daily anomalies ranging from -143 ± 48 to 196 ± 48), while in winter, the diel variation was minimal and close to zero. Together, these patterns in  $\Delta$ /CO<sub>2</sub>, DIC and pH suggest that biological processes, rather than abiotic factors, were the primary driver of their diel variability, with stronger day-night differences during warmer months when both extended daylight hours and higher temperatures enhance photosynthesis and respiration (Regaudie-de-Gioux and Duarte, 2012). This interpretation is further supported by the dominance of the 24-h frequency in the Fourier analysis (Fig. S15), which is characteristic of biologically controlled cycles.


Figure 7. Monthly and daily anomalies of temperature, salinity, TA, DIC, pH and ΔfCO<sub>2</sub>.

Daily anomalies (coloured by day as shown in colorbar) and monthly anomalies cycles (black line) for four representative months: December (winter), May (spring), June (June) and October (Autumn). The monthly means represents the average of all daily cycles for each month. Complete monthly figures are available in the supplementary information (Fig. S7-8, S10, S13, S16, S16, S18, S20).

Figure 8. Monthly and daily anomalies of temperature, salinity, TA, DIC, and pH across tidal phases. Daily anomalies (coloured by day as shown in colorbar) and monthly mean anomalies (black line) for four representative months: March (spring), June (summer), September (autumn) and December (winter). Each subplot shows variations across tidal phases representing the tidal cycle. The monthly means represent the average of all daily tidal cycles for each month. Complete monthly figures are available in the supplementary information (Fig. S9, S12, S14, S19).




## 4.2.3 Combined influence of tidal and biological processes

There were periods across the year when tidal influences on DIC became more pronounced (Fig. S15), particularly in autumn and winter (February, November to January), when biological processes are generally weaker. During these months, DIC anomalies showed minimal day-night variation and followed a semi-diurnal cycle, closely resembling that of TA and water level. This suggests that during the colder months, hydrological processes exerted a stronger influence on DIC fluctuations than biological processes, and this was demonstrated by the 24-h/12-h ratio (Fig. S15), where the tidal component was dominant in winter and autumn. While the influence of these fluctuations on pH was smaller compared to the biologically driven DIC changes, pH monthly anomalies varied slightly, ranging between -0.003  $\pm$  0.001 and 0.004  $\pm$  0.002 (with daily anomalies ranging between -0.04  $\pm$ 0.009 and 0.03  $\pm$  0.008).

Changes in both TA and DIC influence pH, although the underlying drivers differ. While TA was mainly driven by hydrological processes, DIC was driven by both biological and hydrological factors. The relative influence of these processes varied seasonally, with a higher biological influence during warmer months and more hydrological processes during the colder months. To quantify the influence of biological and hydrological processes on DIC, and their subsequent influence on pH, we calculated the Wasserstein distance for the two components of DIC, DIC<sub>bio</sub> and DIC<sub>hydro</sub> which represents the influence of biological and hydrological processes on DIC, respectively. The results (Fig. 9) show a clear seasonal pattern where the Wasserstein distance between DIC<sub>bio</sub> over DIC<sub>hydro</sub> was lower during winter indicating that hydrological processes exerted a stronger influence on DIC during winter period, which was also demonstrated by the Fourier analysis (Fig. S15). Conversely, the greater distance observed for the rest of the year suggests a stronger divergence between biological and hydrological influences on DIC, reinforcing then the observed seasonal differences in TA and DIC variability.

These results are consistent with seasonal scale variations discussed in Section 4.1.2 and highlight that pH is ultimately controlled by biological processes through DIC during most of the year.







# Figure 9. Seasonal variability of the Wasserstein distance between DICbio and DIChydro.

Lower distances indicate stronger hydrological influence, while higher distances indicate more biological influence. The Wasserstein distance is expressed as a percentage using: (Wasserstein<sub>hydro</sub> / (Wasserstein<sub>bio</sub> + Wasserstein<sub>hydro</sub>)) × 100.

#### 4.3 Climate change and the carbonate system

Understanding the carbonate system variability in the Wadden Sea also requires examining large-scale drivers such as climate drivers, that exerts regional to global influences. The North Atlantic Oscillation (NAO) significantly controls weather patterns, ocean circulation and marine biogeochemistry across the North Atlantic region. For instance, a positive NAO index delays spring blooms by 2-3 weeks due to deeper mixed layer depth (Henson et al., 2009), while a negative NAO index weakens the North Atlantic Current transport of subtropical waters, resulting in reduced CO<sub>2</sub> uptake (Thomas et al., 2008). These changes in CO<sub>2</sub> uptake directly affect DIC concentrations and pH in the water column.

The Marsdiep tidal inlet, with its 164-year record of temperature and salinity, offers a valuable dataset to investigate the NAO-driven variability on different time scales. We used temperature and salinity data collected during this study and incorporated long-term data from 2001 until 2024 (Fig. 10). Sea surface temperature in the Marsdiep significantly increased (p-value < 0.005) between 2001 and 2024 at a rate of 0.045 °C per year and was showing a strong correlation with NAO (Pearson's r = 0.60). These changes in temperature, modulated by NAO variability, might have direct consequences for the marine carbonate system. Increasing temperatures result in a reduced CO<sub>2</sub> solubility in seawater and increasing pH levels (Zeebe and Wolf-Gladrow, 2001). Higher temperatures also result in an increased metabolic rate of marine organisms, for instance by influencing carbonate saturation states that are crucial for shell-forming species (Pörtner et al., 2004; Wickins, 1984). During this same period, salinity increased with an average rate of 0.035 per year, most likely due to reduced freshwater discharges into the Wadden Sea (Fig. S21) and increased evaporation (Philippart et al., 2024), contrasting with the observations of Van Aken. (2008a) for the period 1861-2003. Salinity and NAO presented a moderate correlation (Pearson's r = 0.44) between 2001 and 2024, but higher than the correlation found by Van Aken. (2008b). Changes in salinity also influence TA and DIC concentrations, with direct implications for pH.

The influence of NAO on the North Atlantic and North Sea has been extensively researched, however fewer studies have explored its impact on the mixing dynamics between the southern North Sea and the Wadden Sea, particularly regarding the marine carbonate system. The NAO has been shown to strongly affect mixing between the northern and southern North Sea, with a positive NAO index resulting in a higher pH in the northern part and a lower one in the southern part, while a negative NAO index increases mixing between the two regions and results in a more transitional pH pattern (Salt et al., 2013). Since NAO influences the mixing between the northern and southern North Sea, we can then expect a similar effect between the southern North Sea and the Wadden Sea.

Considering that annual pH ranges can vary by 1 in riverine-influenced coastal areas (Blackford and Gilbert, 2007), changes in freshwater inputs, driven by NAO, could amplify this variability in the Wadden Sea. The combination of all these



processes (temperature effect, mixing changes, variable freshwater inputs) suggest that NAO may create an even more complex and non-linear responses in the Wadden Sea and southern North Sea interactions that would warrant further investigation.

Our results provide important insights into the carbonate system in the Marsdiep channel; however, they may not fully represent the entire Wadden Sea system. The Marsdiep channel is the main channel where most of the water exchange occurs between the North Sea and western Wadden Sea, but we must consider regional variations, as conditions vary across the system. The Wadden Sea exhibits a salinity gradient with freshwater influence increasing east. In the northern Wadden Sea, the Ems and Weser rivers contribute to the local variability, whereas the western Wadden Sea is predominantly affected by managed discharges from the Lake Ijssel. These factors, among other factors such as topography, tidal exchange and barrier islands formation, contribute to the spatial heterogeneity of the carbonate system in the Wadden Sea, and may result in different responses to changes driven by NAO across the region. This highlights the need for broader-scale studies across the Wadden Sea, as it is important to understand how climate-driven changes interact with local and regional processes in shaping the carbonate system.

Figure 10. Yearly variability of temperature, salinity and NAO index between 2001-2023. Annual mean temperature (top, red line) and salinity (bottom, green line) alongside NAO index (yearly mean in black line and bars in red and blue for monthly data). The grey shaded area highlights the study period February 2022-January 2023.

### **5 Conclusions**







Due to the combined effect of hydrological, biological, and physical processes, the marine carbonate system in the Wadden Sea exhibits complex processes on different timescales. Seasonal pH variability is primarily driven by biological processes, with maximum pH values observed in spring. TA is mainly influenced by hydrological processes, whereas DIC reflects both hydrological and biological controls. The hydrological effects of TA and DIC on pH approximately cancel each other out, leaving biological processes as the dominant driver of pH. At the diel scale, pH and DIC followed 24-hour cycles that might be linked to NCP and respiration, while TA was mainly driven by the semi-diurnal tidal cycles. While periods of CO<sub>2</sub> uptake occurred during bloom events, the results showed that the Wadden Sea acted as a net CO<sub>2</sub> source to the atmosphere, with an annual release of 4.7 g-C m-2, over the observed period.

We developed an MLR model to predict TA from observed salinity and tidal cycles. The model performance was adequate for seasonal and diel applications. However, due to important spatial and temporal variability in the Wadden Sea, region-specific MLR models may be needed to fully capture TA dynamics across the different basins of the Wadden Sea.

In order to improve our understanding of the drivers of the marine carbonate system across the Wadden Sea, high-resolution time series are essential, ranging from minute-scale for continuous sensor measurements, to weekly or bi-weekly for discrete sampling. This is particularly important given the interplay between tidal cycles, freshwater inputs, and different biological processes which needs to be disentangled in order to assess their relative contribution to marine carbonate chemistry. Such studies are crucial, given the role of the Wadden Sea in alkalinity export towards the North Sea (Norbisrath et al., 2024; Thomas et al., 2009). Furthermore, the contribution of external drivers such as global warming, potential changes in large-scale atmospheric patterns as indicated by the NAO index (Smith et al., 2025) and the changing tides in the Marsdiep channel (van der Molen et al., 2022) may alter the balance between the hydrological and biological controls over time.

Under these changing conditions, it becomes urgent to establish sustained monitoring programs from minutes to weekly timescales in order to resolve the influence of tidal cycles, biological processes and other factors on carbonate chemistry. Understanding the influences of these processes is crucial for distinguishing natural variability from anthropogenic signals in this highly variable system. Only with high-resolution monitoring then we can predict how climate change will affect the Wadden Sea's role in CO<sub>2</sub> uptake, its regional role in alkalinity cycling and the ecosystem. While high-resolution monitoring might require substantial funding, the Wadden Sea's socio-economic and ecological importance (Baer et al., 2017) makes such monitoring essential for informed decision making under changing climate conditions. Understanding the marine carbonate system will support the management of fisheries and aquaculture economies by developing adaptive strategies to different challenges that may arise from ocean acidification.

# Data availability

The dataset presented in this article are freely available online at <a href="https://doi.org/10.25850/nioz/7b.b.4i">https://doi.org/10.25850/nioz/7b.b.4i</a>.

The scripts for processing the data can be found in <a href="https://github.com/YOurradi/jetty">https://github.com/YOurradi/jetty</a> processing by the time of publication.

## 860 Author contribution.

YO, MPH, GJR conceptualized the project. YO, MPH curated the data and performed the investigation. YO conceptualized the methodology, used the necessary software, visualised the data and prepared the original draft of the paper. SvL provided additional observational data. YO, MPH, GJR, SvL reviewed and edited the paper.

### Competing interests.

The authors declare that they have no conflict of interest.

#### Disclaimer.

Publisher's note: Copernicus Publications remains neutral with regard to jurisdictional claims in published maps and institutional affiliations.

# Acknowledgements.

We are grateful to Eric Wagemaakers for installing the pH sensor at the NIOZ jetty and conducting all sensor maintenance. We thank Eric as well as Evaline van Weerlee for providing the hydrological and nutrient data from the NIOZ jetty, including sample collection and data processing which made this work possible. We are thankful to Sharyn Ossebaar for her invaluable assistance and support in the laboratory.

#### 875 Financial support

This research was funded by the Netherlands Enterprise Agency (RVO), project RoboDock (MOOI20FO2JU).

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
