# Peer review of "Marine carbonate system variability from tidal to seasonal timescales at the interface between the North Sea and Wadden Sea"

_EGUsphere, 2025_

## Referee Comment (RC1)

This is an interesting study that applies statistical techniques in new ways to predict high frequency (10 min resolution) changes in TA and DIC in an inlet of the Wadden Sea. The methods are instructive, and include time series kriging (with a semivariogram) for uncertainty analysis, Wasserstein distances to reveal hydrological vs biological influences on DIC, and multiple linear regression for the prediction of total alkalinity. I enjoyed reading about the approach that was used to derive high frequency predictions of TA and DIC from the observations of S, T, and pH.

**Co-reviewer 1:**

The physical-hydrological component of this study could be more clearly described.

- Ridderinkhof et al., (1990, 2002) describe inflow to the Wadden Sea occurring primarily on the south side of the Marsdiep channel and outflow from the Wadden Sea occurring on the North side of the channel - where these measurements were made. Therefore, these data would preferentially show Wadden Sea outflowing waters. How does this alter (if appropriate) your interpretation of results?

- Hoppema (1990, 1993) is cited to suggest that different Wadden Sea waters become more or less dominant at different times. This seems vague. Different water masses are introduced at the end of the paper - beginning at line 812. I'd recommend including clarifying information on the water masses in 4.1.1.

- minor point: 4.1.1, "endmember" seems to be used both correctly - to refer to the riverine and North Sea "source" waters, and incorrectly, at line 514 to refer to the high (TA or salinity) waters observed in the inlet.

- minor point: Line 588, Salt et al., (2014) was cited for showing similar nutrient patterns for the Marsdiep channel, but their study reports pH and CO2 for the North Sea.

Figure 4. Please include Kw, to understand how much of flux is driven by wind.

Figure 4. I found the legend misleading. High frequency TA and DIC were both predicted, not "continuously measured." Perhaps the color used to represent them could be changed to help clarify this.

4.1.3. The ΔTA/ΔDIC discussion is interesting, but it also seems under-developed. Please include references here that could provide more confidence in interpreting these results. How do we know that the winter-to-summer decrease in the ratio isn't simply driven by the reduced rainfall and river input in summer?

Figure 8. It would be helpful to show the tidal height, or to at least indicate the approximate locations of low tide and high tide.

**Co-reviewer 2:**

Fig 1 – the + sign is quite small & hard to see (esp. if printed in black & white), so it would be helpful if you could label the Marsdiep Channel and maybe make the jetty indicator more clear.

Line 120 - The authors say they removed all data that was not collected directly after the cleaning process which happened every 10 minutes. Was there a significant effect of biofouling in the 10 minutes between cleans? Was this necessary?

Line 120 - By placing the equipment at a fixed depth with respect to NAP, the depth of the instruments is changing with respect to the water surface, and the samples are taken at the surface. How can you be sure there is no stratification, which would mean that (a) your sensors are measuring different water masses at different points in the tidal cycle, and (b) your samples are not measuring the same water mass as your sensor? The authors cite Buijsman and Ridderinkhof, 2008 to support the claim that the water column can be assumed to be fully mixed but there is no clear supporting evidence of that in this paper. The other references (Otto 1990, Postma 1954) also don't support this – Otto 1990 explicitly excludes the Wadden Sea in its analysis and Postma 1954 is from over half a century ago. Also, the non-zero coefficient for water level and water level difference in equation (4) (TA-salinity reference) suggests that the water is not fully mixed. If the authors have any data that would better support the assumptions of mixing, that should be included; if not, they will need to address and quantify the uncertainties that this creates.

Section 2.4. – the language is a bit confusing overall here. First, section 2.4.1 is entitled "pH sensor calibration" but there doesn't seem to be any actual calibration happening here, just the implementation of the Nernst equation. Figure 2 shows raw & "corrected" pH sensor data, but it's not clear what the correction process is. It might be that the "uncertainties" calculated in section 2.4.2 were used as corrections, but this is not explicitly stated anywhere in the text and this needs to be clarified and perhaps re-named to reflect that it is a correction process, not a calculation of uncertainty.

Eq. 4 – what is the physical rationale behind having both a sine and a cosine term in the model predicting TA from S?

Line 414 – "In contrast, months with limited sampling (n=1-2), showed unreliable RMSD estimates, as insufficient observations cannot accurately capture the real monthly variability needed by the model to predict TA. These results reflect a sampling limitation rather than model deficiency." I'm not sure that you can meaningfully decouple a sampling vs model weakness, since you're using the samples to create the model.

Line 450 – I'm not sure what is meant by the phrase "However, the measured DIC values exhibited comparable values and patterns" – it sounds like it is comparing DIC_measured to DIC_calc to justify the calculations, but this seems contradictory to the previous two sentences. Note there is a minor typo on line 450, "form" instead of "from".

Abstract - Is it reasonable to make claims about the entire Wadden Sea net CO2 source based on one point measurement?